# Mesenchymal Stem Cells from Familial Alzheimer’s Patients Express MicroRNA Differently

**DOI:** 10.3390/ijms25031580

**Published:** 2024-01-27

**Authors:** Lory J. Rochín-Hernández, Lory S. Rochín-Hernández, Mayte L. Padilla-Cristerna, Andrea Duarte-García, Miguel A. Jiménez-Acosta, María P. Figueroa-Corona, Marco A. Meraz-Ríos

**Affiliations:** 1Departamento de Biomedicina Molecular, Centro de Investigación y de Estudios Avanzados del Instituto Politécnico Nacional, Instituto Politécnico Nacional 2508, Ciudad de México 07360, Mexico; lory.rochinh@cinvestav.mx (L.J.R.-H.); mayte.padilla@cinvestav.mx (M.L.P.-C.); duarte.gandrea@gmail.com (A.D.-G.); miguel.jimenez@cinvestav.mx (M.A.J.-A.); mpfigueroa@cinvestav.mx (M.P.F.-C.); 2Departamento de Biotecnología, Centro de Investigación y de Estudios Avanzados del Instituto Politécnico Nacional, Instituto Politécnico Nacional 2508, Ciudad de México 07360, Mexico; lory.rochin@cinvestav.mx

**Keywords:** miRNA, transcriptome, Alzheimer’s disease, familial Alzheimer’s disease, Jalisco mutation, PSEN1, A431E, mesenchymal stem cells, olfactory, biomarkers, epigenetics

## Abstract

Alzheimer’s disease (AD) is a progressive neurodegenerative disorder and the predominant form of dementia globally. No reliable diagnostic, predictive techniques, or curative interventions are available. MicroRNAs (miRNAs) are vital to controlling gene expression, making them valuable biomarkers for diagnosis and prognosis. This study examines the transcriptome of olfactory ecto-mesenchymal stem cells (MSCs) derived from individuals with the PSEN1(A431E) mutation (Jalisco mutation). The aim is to determine whether this mutation affects the transcriptome and expression profile of miRNAs and their target genes at different stages of asymptomatic, presymptomatic, and symptomatic conditions. Expression microarrays compare the MSCs from mutation carriers with those from healthy donors. The results indicate a distinct variation in the expression of miRNAs and mRNAs among different symptomatologic groups and between individuals with the mutation. Using bioinformatics tools allows us to identify target genes for miRNAs, which in turn affect various biological processes and pathways. These include the cell cycle, senescence, transcription, and pathways involved in regulating the pluripotency of stem cells. These processes are closely linked to inter- and intracellular communication, vital for cellular functioning. These findings can enhance our comprehension and monitoring of the disease’s physiological processes, identify new disorder indicators, and develop innovative treatments and diagnostic tools for preventing or treating AD.

## 1. Introduction

Alzheimer’s disease (AD) is a degenerative neurological disorder that gradually worsens over time and is the most common type of dementia globally. The escalating prevalence of AD poses a burgeoning public health concern, especially given the expanding elderly population and the regrettable absence of efficacious approaches for diagnosis, predicting the disease’s advancement, or offering a remedy [1].

AD can be classified into two primary groups according to the commencement of symptoms. About 95% of cases of Alzheimer’s disease occur in individuals who are over the age of 65. This form of the disease is known as late-onset Alzheimer’s disease (LOAD) or sporadic Alzheimer’s disease (SAD). The remaining cohort manifests symptoms of early-onset Alzheimer’s disease (EOAD) before reaching the age of 65. Among the cases in this group, familial Alzheimer’s disease (FAD) accounts for approximately 80% and is triggered by mutations in three primary genes: Presenilin-1 (*PSEN1*), Presenilin-2 (*PSEN2*), and the amyloid precursor protein (*APP*) gene [2]. Both groups share pathological characteristics, including the presence of amyloid plaques composed of amyloid β (Aβ) peptides and neurofibrillary tangles (NFTs) made up of hyperphosphorylated tau proteins. They also display similar clinical symptoms, such as progressive memory loss and cognitive decline, that ultimately result in dementia [3]; over 20% of FAD patients can develop atypical symptoms like spastic paraparesis, dysarthria, schizophrenia, Parkinsonism, and depression [4,5,6]. Familial Alzheimer’s disease (FAD) offers a distinctive biological model for comprehending the underlying mechanisms of neurodegeneration in Alzheimer’s disease (AD). This is attributed to the advancement of animal and cellular models established using the precise mutations accountable for the disease’s onset. A specific PSEN1 mutation, the PSEN1(A431E) mutation, has been identified in Mexico. This mutation is associated with familial Alzheimer’s disease and has a founder origin in the state of Jalisco. It is worth noting that over 500 mutations in these three genes are linked to familial Alzheimer’s disease [7,8].

The advancement of “omic” sciences, employing cutting-edge techniques like next-generation sequencing, mass spectrometry, microarrays, and deep learning methods, enables us to analyze multi-omics data and gain a holistic and thorough comprehension of diseases like Alzheimer’s disease (AD) [9,10]. Current research has highlighted the importance of epigenetics in the progression of Alzheimer’s disease (AD). This study focuses on investigating the influence of DNA methylation, chromatin remodeling, histone modifications, and the control of non-coding RNAs in individuals with Alzheimer’s disease (AD), as well as in animal and cellular models that are relevant to this condition [11,12]. 

RNA can be categorized into two main types: messenger RNAs (mRNAs), which encode proteins, and non-coding RNAs (ncRNAs), which do not encode proteins but can function as regulatory factors during transcription or post-transcription. ncRNAs are classified into two major categories: small non-coding RNAs, which are less than 200 nucleotides in length, and long non-coding RNAs, which are greater than 200 nucleotides in length. Examples of small non-coding RNAs encompass ribosomal RNAs (rRNAs), transfer RNAs (tRNAs), small nuclear RNAs (snRNAs), small nucleolar RNAs (snoRNAs), and microRNAs (miRNAs), among various others [13].

The current study evaluates transcriptomic assays using the GeneChip^®^ Human Gene 2.0 ST microarrays (Affymetrix, Santa Clara, CA, USA). Nevertheless, our study concentrates on the miRNA profiles and their corresponding target genes. miRNAs are short RNA molecules consisting of about 22 nucleotides. They are well-known for their crucial function in controlling gene expression. Their biogenesis entails a series of sequential steps wherein the primary transcript undergoes processing to generate the mature miRNA, transitioning from the nucleus to the cytoplasm [14]. MiRNAs are primarily responsible for suppressing gene expression after transcription through the RNA-induced silencing complex (RISC). Although miRNAs are known mainly for suppressing protein expression by degrading or inhibiting mRNA translation, they can also participate in various unconventional regulatory mechanisms. These include encoding peptides from pri-miRNAs, which increases the expression of the corresponding mature miRNAs.

Additionally, miRNAs can interact with RNA-binding proteins other than Argonaut, acting as decoys and interfering with their function. They can also serve as ligands for Toll-like receptors (TLRs), initiating the signaling cascade of the inflammatory response or causing neurodegeneration. Furthermore, miRNAs can modulate transcription and translation, among other functions [15].

Most currently known miRNAs have been discovered in the nervous system. Studies using animal models have demonstrated that the absence of these miRNAs can significantly affect both normal and abnormal brain functions [16]. Indeed, changes in miRNA expression have been detected in the brains of individuals with Alzheimer’s disease and different biological fluids, including serum, plasma, and cerebrospinal fluid (CSF) [17]. Nevertheless, few investigations have investigated the miRNA expression patterns in mutations associated with familial Alzheimer’s disease (FAD). One such study utilized microarrays to analyze the cerebrospinal fluid (CSF) of four individuals from a Chinese family who carried the G378E mutation in PSEN1 protein [18]. The second study, conducted by Fung et al., utilized a murine model with the N141I mutation in the PSEN2 protein. They generated induced-pluripotent stem cells (iPSCs) from fibroblasts of individuals carrying this mutation and subsequently differentiated them into microglia [19]. 

Although the brain is the central tissue examined in research on Alzheimer’s disease (AD), our comprehension of the disease’s initial phases is restricted due to its inaccessibility in living patients [20]. Most studies examining cells of the neural lineage of the disease focus on cells that have already undergone differentiation, such as neurons and microglia [21,22,23]. A limited amount of research assesses the deregulation of miRNAs in neural precursor cells. A potential source of neural stem cells is the olfactory mucosa, where neurogenesis is necessary to replace the olfactory neurons. Ecto-mesenchymal stem cells (MSCs) show great potential because they can be obtained from the olfactory nasal niche without invasive procedures [24]. According to Benítez King’s group, the cells in passage six exhibit a uniform population of MSCs under the specific medium conditions.

Furthermore, multiple research groups have extensively investigated the properties and characteristics of these cells, such as differentiation, adherence, and clonogenicity [24,25,26]. These cells exhibit a higher neurogenic potential than mesodermal lineage due to their ectodermic origin, making them valuable for treating neurodegenerative disorders. This particularity is significant because many patients with Alzheimer’s experience a reduced sense of smell years before symptoms appear [24,27,28]. Moreover, Aβ and tau protein aggregates have been detected through all the olfactory pathways, the olfactory bulb, and the neuroepithelium [29,30,31,32]. Studies have also shown an altered neurogenesis, with a lower viability of olfactory neurons compared to controls [33]. Our group recently published a study on the modified proteostasis network and characterized MSCs obtained from individuals carrying the PSEN1(A431E) mutation before and after developing symptoms. We used two label-free proteomic methods to analyze the data [34]. Currently, there is a lack of studies investigating the dysregulation of transcriptome or miRNA expression levels and their target genes in this specific cellular type carrying the PSEN1(A431E) mutation. This study examines the transcriptome of MSCs derived from individuals with the PSEN1(A431E) mutation who are asymptomatic, presymptomatic, or symptomatic. The goal is to determine if this mutation, associated with neurodegeneration, impacts the transcriptome and miRNA expression profiles. This study aims to understand how these changes in gene expression relate to the cellular physiology of affected cells and their correlation with disease stage. Ultimately, this study seeks to identify new biomarkers for Alzheimer’s disease.

## 2. Results

RNA was extracted from olfactory ecto-mesenchymal stem cells obtained from four Mexican-mestizo family members who carried the PSEN1(A431E) mutation (Figure 1). Additionally, RNA was extracted from four healthy control subjects of a similar age, called C15, C35, C42, and C55. Individuals carrying the PSEN1(A431E) mutation manifest symptoms at approximately 43 years of age. The clinical histories, neurological and cognitive information of this family, which has a well-documented three-generation history of spastic paraparesis, were described by Santos-Mandujano. We recently published the proteomic profile of cells obtained from individuals carrying the PSEN1(A431E) mutation before and after symptoms appeared and compared them to their respective control groups [34]. The presymptomatic carrier (P44) was a 44-year-old woman with normal cognitive function, but who was experiencing mild leg fatigue, increased reflexes throughout the body, and a slight decrease in the sense of smell.

On the other hand, the symptomatic carrier (P54) was a 54-year-old man who had been experiencing lower extremity motor impairment for seven years, minor cognitive difficulties, typical symptoms of upper motor neuron disease, and a loss of sense of smell. Furthermore, this study incorporated two male individuals, aged 18 (P18) and 36 (P36), who carried a mutation but had not yet exhibited any disease symptoms. These individuals were considered asymptomatic carriers. The purpose was to investigate whether the transcriptome of their MSCs displayed any early-stage abnormalities. 

### 2.1. RNA Quality and Integrity

After extraction, we evaluated the RNA quality and integrity in each sample used for the microarrays by measuring the RNA integrity number (RIN) using a Bioanalyzer 2100 (Agilent, Santa Clara, CA, USA). Samples with an RIN value of 8 or higher were only included to guarantee exceptional quality and integrity. Figure 2 displays the electropherograms containing these specific values. 

### 2.2. Altered Transcriptomic Profile of MSCs Derived from PSEN1(A431E) Mutation Carriers by Disease Stage

After confirming the quality of the RNA, our objective was to examine whether there were distinct miRNA patterns based on age or disease stage in individuals with the PSEN1(A431E) mutation. To accomplish this, we utilized GeneChip Human Gene 2.0 ST microarrays to compare each carrier with a control of the same age. An analysis of variance (ANOVA) was employed to produce a list of genes that exhibit significant differences between MSCs harboring the PSEN1(A431E) mutation and control cells. The criteria for inclusion in the list were a fold change greater than two or less than −2 and a *p*-value lower than 0.05, adjusted for a false discovery rate (FDR) of less than 0.05. After applying statistical criteria, transcripts were filtered and selected from the microarray database (refer to the Section 4). This process resulted in identifying miRNAs and mRNAs that exhibited differential expression at each symptomatologic stage. Only mRNAs and miRNAs that showed differential expression at each symptomatologic stage were chosen (Figure 3A). In the case of the symptomatic stage (S), data mining resulted in 1992 total transcripts, of which 1581 corresponded to coding transcripts (mRNAs), including 75 to miRNAs, 251 to lncRNAs, 71 to snoRNAs, 10 to snRNAs, 3 to vtRNAs, and 1 to a component of RNA telomerase (Figure 3B and Appendix A). Regarding the presymptomatic stage (P), there were 1772 transcripts in total, of which 1376 were mRNAs, 79 miRNAs, 282 lncRNAs, 31 snoRNAs, 3 snRNAs, and 1 was a vtRNA (Figure 3C and Appendix A). In the asymptomatic stage, 903 transcripts were obtained from the 30-year-old group (A3). Of these, 603 were mRNAs, 81 were miRNAs, 193 were lncRNAs, 24 were snoRNAs, 1 was an snRNA, and 1 was a vtRNA (Figure 3D and Appendix A). In the asymptomatic 20-year-old group (A2), we obtained 1681 transcripts. Of these, 1350 were mRNAs, 97 were miRNAs, 108 were lncRNAs, 121 were snoRNAs, 4 were snRNAs, and 1 was a vtRNA (Figure 3E and Appendix A). The transcripts that were expressed differentially (DETs) varied across patient categories based on age and symptoms. The number of DETs was similar across categories, ranging from 1300 to 1500, except for asymptomatic individuals in their 30s with 603 DETs. Due to the limited number of samples analyzed, it was challenging to establish a definitive correlation between the number of differentially expressed transcripts (DETs) and the stage of the disease or mutational status. Thus, our results required validation. However, our statistical criteria were more stringent than other transcriptomic studies in AD.

As anticipated, the carrier displaying symptoms exhibited the highest count of differentially expressed transcripts (DETs) (Figure 3B). However, intriguingly, the asymptomatic youngest carrier also demonstrated a significant number of DETs and the highest count of altered miRNAs (Figure 3E). These changes implied substantial alterations in the initial stages due to the mutation.

### 2.3. miRNAs Profile of MSCs Derived from PSEN1(A431E) Mutation Carriers by Disease Stage

Figure 4A–D display the variation in miRNA expression according to different stages. In the symptomatic stage, the miRNAs miR-548F5 and miR-548C showed a more pronounced increase in expression with FCs of 26.95 and 11.55, respectively, while miR-1178 and miR-924 showed a decrease in expression (FC = −6.85 and −4.92) (Figure 4A). miR-3142, miR-4635, and miR-130B exhibited the highest overexpression levels during the presymptomatic stage (FC = 5.41, 5.33, and 4.97), while miR-4521 (FC = −9.19) and miR-1244-1/2/3/4 (FC = −6.81) were under expressed, as shown in Figure 4B. In the asymptomatic stage, the group of individuals aged 30 emphasized the upregulation of miRNAs miR-4660 (FC = 6.39) and miR-4293 (FC = 4.06) and the downregulation of miR-3189 (FC = −5.48) and miR-4799 (FC = −4.8) (Figure 4C). In the 20-year-old group, the expression levels of miR-543 (FC = 7.16), miR-376B (FC = 5.73), and miR-154 (FC = −4.05) were found to be higher than usual, while the expression levels of miR-1234 (FC = −4.65), miR-143 and miR-145 (both with FC = −4.05) were found to be lower than expected (Figure 4D). 

Interestingly, asymptomatic stages present a more significant number of overexpressed miRNAs. In contrast, there is a greater number of under expressed miRNAs in the presymptomatic and symptomatic stages. The quantity of under- and overexpressed miRNAs is very similar. This may suggest that compensatory mechanisms are underway from the early stages of the illness.

### 2.4. Target Interactions and Genes Affected by Altered miRNAs in Individuals with the PSEN1(A431E) Mutation According to the Stage of the Disease

After identifying the modified miRNAs according to their stage, we searched to identify their targets and determine the potential metabolic pathways and biological processes that were affected. We utilized a total of 5 databases for this purpose. The putative targets were searched using the TargetScan, miRDB, and DIANA-microT databases. Nevertheless, this query could have yielded many outcomes, and not all were likely to be authentic. As a result, we specifically chose the common targets found in all three databases for the putative genes (Figure 5, Appendix A).

In addition, we referred to TarBase and miRTarBase to acquire data regarding targets that have been experimentally validated (Appendix A). It is widely recognized that each microRNA (miRNA) can regulate multiple genes and vice versa. Multiple miRNAs can control a single gene. Figure 5A exhibits a comprehensive table that showcases all the identified interactions involving the modified miRNAs in each database. 

To determine the genes most strongly regulated by miRNAs during each stage of symptom development, we created a compilation of the top 10 genes. The regulation of these genes is influenced by more miRNAs, as evidenced by both putative and experimentally confirmed interactions (Figure 5B). The molecules *WEE1*, *BTG2*, *DICER1*, and *ANKRD52* were significantly identified as being standard to all four conditions. Many of these genes, which undergo increased regulatory pressure, notably impact the cell cycle, cell growth, and proliferation. The analysis of our findings identified changes in a limited number of genes. Out of the genes exhibiting the highest regulatory pressure as determined by the databases (top 10 predicted and validated) in the mRNA results of the microarrays, these four genes were found to be modified: *PTEN*, *CDK6*, *CDKN1A*, and *CCND1*. The symptomatic carrier exhibited *PTEN*, *CDK6*, and *CDKN1A* overexpression with fold changes (FCs) of 2.01, 4.06, and 3.03, respectively. CDK6 exhibited underexpression with a fold change (FC) of −2.58 in the presymptomatic carrier, whereas CDKN1A and *CCND1* showed overexpression with FCs of 3.22 and 2.58, respectively. 

Following the elimination of duplicate entries arising from the multiple interactions between targets and different miRNAs, the unique target genes from putative and validated databases were employed for subsequent bioinformatic analyses. The findings of these analyses are displayed in Figure 5C.

Remarkably, the cells of the youngest asymptomatic carrier, 20 years old (A2), exhibited the most significant quantity of modified miRNAs. Additionally, this individual also displayed the highest number of interactions and mRNAs. 

We utilized Venn diagrams to identify the targets overlapping the putative, validated, and altered targets in MSCs carrying the PSEN1(A431E) mutation. We selected targets that coincided with our findings and had been validated in different biological samples (intersection highlighted in Figure 6). Please refer to Figure 6 and Appendix A for further details. Notably, over 60% of the transcripts deregulated in the mutation carriers, except for the asymptomatic case A3, could be regulated by the altered miRNAs during the disease progression. Some 75.8% (1199) of the 1581 transcripts led to the symptomatic stage (Figure 6A), while for the presymptomatic case, 76.2% (1048/1376) resulted in this stage (Figure 6B). In the asymptomatic cases, 48.4% (284/603) and 66.7% (901/1350) of the transcripts led to stages A3 and A2, respectively (Figure 6C,D). Validating the putative targets in our MSCs could enhance the regulatory network for these miRNAs in Alzheimer’s disease, which has not been reported previously.

### 2.5. The Gene Ontology and Signaling Pathways Are Disrupted in Individuals Carrying the PSEN1(A431E) Mutation at Different Stages of the Disease

The WebGestalt tool was used to analyze the effect of miRNA dysregulation on biological mechanisms and metabolic pathways at each symptomatologic stage. The utilization of Venn diagrams allowed for the visualization of the connections between the target putative genes, the validated genes, and the unique pathways of our cellular model, which correspond to the pathways identified in the mRNA_A431E (Appendix A). Appendix A presents the 30 most significant biological processes (BPs), molecular functions (MFs), and cellular components (CCs).

The symptomatic and presymptomatic conditions were characterized by biological processes predominantly associated with DNA replication, cellular division, and the cell cycle. Eight biological processes were familiar to the altered PSEN1(A431E) mutation, and transcripts were validated in symptomatic conditions. On the other hand, six were shared in the condition where there were no symptoms yet, and these were mainly associated with processes that control the cell cycle. During asymptomatic conditions, the biological processes primarily involve cell adhesion and the synthesis and release of cytokines in A3. In A2, their leading associations are interferon response, cell division, and the cell cycle (Appendix A).

The symptomatic data showed that the cell–substrate junction, PML body, and Golgi-associated vesicle were common CCs used to validate and alter PSEN1(A431E) transcripts. In contrast, the asymptomatic 30s only had an enrichment of the extracellular matrix as a cellular component. However, all conditions of altered PSEN1(A431E) mRNAs showed an enrichment of the extracellular matrix. Growth factor binding, helicase activity, and cell adhesion molecule binding were identified as significant molecular functions in the symptomatic stage. In the presymptomatic condition, cyclin-dependent protein kinase activity and extracellular matrix structural constituent were critical, with the latter also playing a crucial role in the asymptomatic A2 stage (Appendix A). 

Enriched pathways were obtained for each stage of symptoms using the KEGG database. Figure 7 displays Venn diagrams illustrating the resulting signaling pathways in the three datasets, categorized by symptom stage (Appendix A). The presence of symptoms and early symptoms resulted in a more significant number of signaling pathways being modified. The cellular senescence and p53 signaling pathways were found in all three datasets of the symptomatic condition. These two pathways were also present in the four stages of symptomatology. The cell cycle pathway was observed in both the symptomatic and presymptomatic stages.

In contrast, DNA replication, valine, leucine, and isoleucine degradation, and necroptosis were exclusively present in the symptomatic stage. The presymptomatic stage was crucial to the lysosome, FoxO, and NF/kappa B signaling pathways. The Toll-like receptor and PI3K-Akt signaling pathways were more abundant in the asymptomatic A3 stage. In the A2 stage, pathways related to antigen processing and presentation, cell adhesion molecules (CAMs), and NOD-like receptors were enriched.

In addition, datasets were analyzed individually with REACTOME, PANTHER, and Wikipathways databases and in the WebGestalt tool (Appendix A). Multiple pathways coincided in these databases, principally involved in cell cycle phases, cell division, proliferation, cellular senescence, cancer-related pathways, and cytokine and interferon signaling pathways. 

### 2.6. PSEN1(A431E) Mutation Transcriptomic Profile 

While the previous findings demonstrated distinct transcriptomic profiles associated with different stages of symptoms, they also exhibited certain commonalities. We sought to determine whether there was a distinct transcriptomic profile in MSCs harboring the PSEN1(A431E) mutation compared to cells lacking the mutation. To gain a comprehensive understanding of the data, we conducted principal components analysis (PCA) (Figure 8A). Mutation is the primary cause of variation with age. In the control group, the expression profile remained consistent regardless of the age of the individuals from whom the cells were obtained. The disparities in the manifestation were verified via hierarchical clustering displayed in a heat map, enabling us to distinguish between cells derived from A431E mutation carriers and controls, irrespective of the stage. The diagram is depicted in Figure 8B. To accomplish this, we combined the four datasets containing modified transcripts for each stage and calculated the average expression of the PSEN1(A431E) mutation carriers and the average of the controls. We then considered a fold change (FC) of ≥1.5 or ≤−1.5 for the modified transcripts. The data yielded 1241 mRNAs and 81 miRNAs, with 36 being downregulated and 45 upregulated (Appendix A).

### 2.7. PSEN1(A431E) Mutation miRNA Profile 

We performed a Venn diagram analysis to determine if MSCs with the PSEN1(A431E) mutation exhibited a distinct miRNA profile. We included all the miRNAs that were altered in different symptomatic stages (Figure 9A) and selected those that were changed in at least two carriers of the PSEN1(A431E) mutation (54 miRNAs). We then compared these miRNAs with the 81 miRNAs selected by FC, resulting in 29 miRNAs (Figure 9B). Among these, 12 were downregulated, and 17 were upregulated. Among the miRNAs analyzed, 8 showed the highest fold change (FC) and were found to be altered in at least two individuals with mutations (Figure 9C). miR-130B was the only miRNA shared among all the symptomatic conditions and had the highest FC of 4.1 (Figure 9C).

When considering a fold-change (FC) value greater than or equal to 1.5, it was observed that five miRNAs, namely miR-147B, miR-3142, miR-4429, miR-146A, and miR-4540, were consistently upregulated across all stages. Figure 9B shows an intriguing pattern in the last one, where it is initially overexpressed but gradually decreases as the disease progresses.

Furthermore, the miRNAs miR-4521, miR-1178, miR-4668, and miR-32 exhibited significant downregulation across all conditions, as indicated by their consistently low fold-change values (FC ≤ −1.5) (Figure 9C). The expression levels of miR-3614 and miR-1244 were also consistently reduced in all experimental conditions, as shown in Figure 9B.

Collectively, miR-143, miR-145, miR-4438, and miR-4677 exhibited a decrease in expression. However, intriguingly, this decrease was observed in all conditions except for the symptomatic carrier. In contrast, miR-4635, miR-548AB, and miR-4518 exhibited increased expression in all conditions except the symptomatic carrier (Figure 9B). The results indicate a clear miRNA profile in the MSCs of individuals with the PSEN1(A431E) mutation. 

### 2.8. Functional Analysis of mRNAs in PSEN1(A431E) Mutation Carriers vs. Controls

Subsequently, the WebGestalt tool was utilized to conduct gene ontology and KEGG enrichment analysis to ascertain the specific processes and pathways associated with the 1241 modified mRNAs (Appendix A). Primary biological processes are associated with the cell cycle and its regulation and the response to type I interferon (Figure 10A). The cellular components that showed enrichment were the nucleus, midbody, and extracellular matrix (Figure 10B). The enriched molecular functions included DNA secondary structure binding, extracellular matrix binding, and glycosaminoglycan binding (Figure 10C). The analysis showed an enrichment of pathways associated with the cell cycle, p53 signaling, and cellular senescence, as indicated in Figure 10D.

Furthermore, we conducted a Venn diagram analysis to identify the altered KEGG pathways in the four symptomatologic stages. Among the enriched pathways that were shared by all the mutation carriers, there were 53 signaling pathways, including cell cycles, longevity regulating pathways, TNF signaling pathway, protein processing in the endoplasmic reticulum, autophagy, endocytosis, glycolysis/gluconeogenesis, the calcium signaling pathway, lysosome, and others (Figure 10E). The symptomatic and presymptomatic stages share common processes such as glycosaminoglycan biosynthesis, proteasome, and spliceosome. 

PI3k/Akt signaling pathway and apoptosis were found to be shared in the asymptomatic cases (Appendix A). Furthermore, using the DAVID tool, we employ the Genetic Association Database (GAD) as a resource for genetic disease mapping. The GAD aims to gather, standardize, and store data derived from genetic association studies. The findings indicated that the primary categories of related illnesses encompass the terms cancer, aging, kidney, and neurological disease. Among the genes identified, 16 were associated with Alzheimer’s Disease. These genes include *CXCL8*, *BCHE*, *CEP55*, *CDK1*, *IFIT1*, *IFIT3*, *IL1R1*, *IL6*, *KIF11*, *MKI67*, *MAP3K8*, *PTGS2*, *PPP1R3C*, *TET1*, *TDRD1*, and *VCAM1*.

### 2.9. Comparision between miRNAs Altered in MSCs Derived from PSEN1(A431E) Mutation Carriers and Studies Previously Reported

Ultimately, we aimed to determine whether the miRNAs we obtained were previously linked to Alzheimer’s disease (AD). To accomplish this, we thoroughly searched the 272 miRNAs altered in the PSEN1(A431E) mutation carriers (as depicted in Figure 9A). Our objective was to identify the existing literature that associated these specific miRNAs with Alzheimer’s. Figure 11 displays the miRNAs reported in more than four papers, which were altered explicitly in certain stages of the disease, and those that changed in more than two stages in relation to AD. Appendix A includes the complete list of references for the tables in Figure 11. It is well established that various studies can present distinct findings regarding the miRNAs affected in Alzheimer’s disease (AD). As a direct search in PubMed does not produce these results, conducting a comprehensive search is necessary. Nevertheless, this direct search provides a rough understanding of AD’s most extensively researched miRNAs.

## 3. Discussion

Recent studies have shown that miRNAs control approximately 70% of the human genome, with the brain expressing more than 70% of these miRNAs [35]. Hence, the modification of miRNAs could potentially have a significant impact on the progression of neurodegenerative disorders. So far, numerous miRNAs have been identified as being changed and playing roles in controlling the formation and removal of protein aggregates. This alteration has a notable effect on the development of various neurodegenerative diseases such as Alzheimer’s disease (AD), Parkinson’s disease (PD), Huntington’s disease (HD), and amyotrophic lateral sclerosis (ALS) [36]. In AD, miRNAs in serum, cerebrospinal fluid, and brain tissue, as well as in transgenic models, have been studied [37,38,39,40,41]. Few studies have reported miRNAs in the early stages or related to the progression of AD [22,41,42,43,44,45,46]. Nevertheless, the specific alterations in the transcriptome of mesenchymal stem cells (MSCs) obtained from individuals with familial Alzheimer’s disease (FAD) are not yet understood at various stages of the condition. Therefore, we utilized microarrays to identify distinct miRNA profiles in individuals carrying the PSEN1(A431E) mutation at various stages of the disease.

### 3.1. miRNAs Altered in MSCs Derived from the PSEN1(A431E) Symptomatic Mutation Carrier

During the symptomatic stage, changes were observed in 75 miRNAs, with 39 showing under expression and 36 showing overexpression. The miR-548F5 and miR-548C exhibited the highest levels of overexpression, with fold changes (FC) of 26.95 and 11.55, respectively. Notably, miR-548C decreased in asymptomatic stages (A2 and A3), and miR-548F5 increased in the A3 condition. The miR-548 family, consisting of 69 members, plays significant roles in various biological processes. These processes include regulating the actin cytoskeleton, the MAPK signaling pathway, ubiquitin-mediated proteolysis, cell cycle, axon guidance, and several human diseases such as colorectal cancer, glioma, and AD [47]. In addition, the members of the miR548 family suppress the growth, movement, and infiltration of different types of cells, such as glioma, melanoma, and osteoblast-like cells, by reducing the expression of *c-Myb* [48], *HMGB1* [49] *MAFB*, and *STAT1* [50], genes already reported as being associated with AD and neuroinflammation [51,52,53,54]. Also, overexpression of miR-548k in breast cancer cells is associated with decreased *ABCG2* gene expression, another key player in AD [55,56]. Also, in a bioinformatic analysis previously reported by Ma et al., miR-548C-3p was proposed as a critical miRNA in AD since it was found to be a common target for *CITED2*, *GABRA2*, and *ADAMTS1*, key enzymes in APP processing and extracellular matrix remodeling [57]. Furthermore, it is essential to highlight that different members of the miR548 family are modified depending on the stage. In symptomatic cases, miR-548I2 is increased, while in presymptomatic conditions, miR-548AN, miR-548H2, and miR-548O2 are decreased, and miR-548AJ1 is increased. In the asymptomatic stage A3, the expression levels of miR-548A3, miR-548AK, and miR-548I4 are abnormal, while miR-548A2, miR-548V, and miR-548I are modified in the A2 condition.

Of additional note is that the miRNAs found to be upregulated in the symptomatic carrier and which have previously been reported to be associated with Alzheimer’s disease include miR-181A2 and miR-328. The expression of miR-181A was increased in individuals diagnosed with mild cognitive impairment (MCI) who subsequently developed Alzheimer’s disease (AD) [58]. However, it was downregulated in extracellular vesicles of CSF in AD patients [59]. Furthermore, miR-181a was observed to be significantly elevated in the hippocampus of 3xTg-AD mice. It acts as a negative regulator for synaptic plasticity, facilitates Aβ-induced synaptotoxicity, and inhibits the protein expression of the CREB1 transcription factor [60]. Several studies have reported that miR-328 is involved in the pathophysiology of AD. It is sub-expressed in the brain of AD patients and the APP/PS1 rat hippocampus and regulates *BACE1* [61,62]. BACE1 overexpression attenuates the effect of miR-328 overexpression on neuron injury and inflammation in cellular AD models [63].

The miRNAs miR-1178 and miR-924 showed the most significant downregulation among the symptomatic carriers, with fold changes of −6.85 and −4.92, respectively. The expression of miR-1178 was decreased in the brains of patients with Alzheimer’s disease [64]. Unlike in miR-548, it makes sense that miR-1178 has the opposite function, promoting proliferation, G1/S transition migration, and invasion while inhibiting apoptosis in pancreatic cancer cells, esophageal squamous carcinoma cells (ESCC), nasopharyngeal carcinoma and bladder cancer cells by targeting CHIP, a co-chaperone member of E3 ubiquitin ligase that promotes the ubiquitination and degradation of numerous proteins, affecting AADAC, STK4 and p21 expression [65,66,67,68]. Furthermore, miR-1178 is similarly reduced in the other PSEN1(A431E) conditions (with a fold change of −1.5 or less), as observed in the prefrontal cortex of patients with Alzheimer’s disease. [22]. In contrast, miR-924 is a tumor suppressor in hepatocarcinoma cells and lung cancer [69,70], which inhibits the RHBDD1/Wnt/β-catenin signaling pathway [71]. Moreover, the ectopic expression of lncRNA n335586 significantly decreases the miR-924 and enhances the expression of *CKMTIA*, promoting cell migration/invasion and EMT progression in HCC cells [69].

The microRNAs miR-155, miR-21, miR-29A, miR-193B, and miR-221 have been consistently associated with Alzheimer’s disease (AD) and were also observed to be downregulated in the symptomatic carrier. miR-155 has been extensively researched as a pro-inflammatory microRNA crucial in developing and advancing neurological disorders [72]. In contrast to our findings, previous studies have reported an upregulation of miR-155 in the brains of 3xTg AD mice and the brains of patients with AD [73,74], and also that its overexpression promoted Aβ production and neurofibrillary development [75]. However, miR-155 was downregulated in AD neurons, and the exosome levels were upregulated [76]. In addition, Wang et al. found that the activation of Notch1 in endothelial cells derived from bone marrow inhibited the production of miR-155 [77]. This research is interesting since Notch is a γ-secretase substrate. It was also reported that PSEN1 L435F mutation caused a gain of function in human iPSC-derived 3D cortical spheroids and increased Aβ43 levels [78]. This peptide also increased in the symptomatic PSEN1(A431E) mutation carrier. A similar phenomenon was observed with miR-21, as β-catenin, an essential substrate of γ-secretase, regulated miR-21 expression through *STAT3* and promoted invasion in glioma cells [79]. The expression of miR-29A was reduced in extracellular vesicles in the plasma and serum of patients diagnosed with Alzheimer’s [59,80]. Nevertheless, it was excessively expressed in neurons derived from induced pluripotent stem cells (iPSCs) that carried the PSEN1ΔE9 deletion and in exosomes derived from cells that carried the Swedish mutation [81]. However, it was discovered that there was an excessive expression of this protein in neurons that were derived from induced pluripotent stem cells (iPSCs) with the PSEN1ΔE9 deletion. Exosomes derived from cells with the Swedish mutation also showed high protein levels [82,83,84,85]. miR-221 was downregulated in the blood of patients with Alzheimer’s disease (AD), and it acted as a negative regulator of *ADAM10*. Suppressing its activity resulted in an elevation of ADAM10 levels in SH.SY5Y cells [86,87].

### 3.2. miRNAs Altered in MSCs Derived from the PSEN1(A431E) Presymptomatic Mutation Carrier

During the presymptomatic stage, a total of 79 miRNAs exhibited deregulation, with over 50% of them being downregulated. Specifically, 49 miRNAs were downregulated, including miR-4521 (fold change = −9.19) and miR-1244-1/2/3/4 (fold change = −6.81), which showed the most significant downregulation. Remarkably, miR-4521 exhibited downregulation in all conditions, with a fold change (FC) of ≤−1.5. While there are no specific references linking AD to these miRNAs, it has been reported that they are downregulated in various types of cancer, thereby promoting cell proliferation and invasion. For instance, in the case of breast cancer, the expression of miR-4521 results in elevated levels of reactive oxygen species (ROS) and the occurrence of DNA damage in cells [88]. In hepatocarcinoma cells (HCC), downregulation of miR-4521 increases cell proliferation and invasion via the phosphorylation of the FAK/AKT pathway through FAM129A upregulation [89]. Similarly, in osteosarcoma cells, miR-1244 inhibits viability and proliferation [90]. In ovarian cancer cells, the inhibition of miR-1244 promotes proliferation and aerobic glycolysis (*PKM2*, *HK2*, and *PDK1*) and *ROCK1* expression [91,92]. Additional miRNAs that have been found to be downregulated in relation to Alzheimer’s disease (AD) include miR-125B. This particular miRNA has been shown to contribute to the excessive phosphorylation of tau protein by increasing the expression and activity of kinases, such as Erk1/2 and p35, an activator of the kinase cdk5. Furthermore, miR-125B downregulates the production of phosphatases, including *DUSP6* and *PPP1CA* [93]. It is downregulated in the plasma EV of AD patients [94], and it has been reported as being upregulated in the CSF, brains, and EVs of blood from AD patients [59,93,95,96,97,98,99]. miR-27B is also increased in plasma extracellular vesicles, monocytes derived from blood, and brain tissue of patients with Alzheimer’s [84,96,100], and it is downregulated in plasma of AD patients [42]. This pro-inflammatory microRNA triggers the activation of microglia and enhances the production of TNFα, IL-6, and IL-1β, as well as apoptosis [101]. Moreover, silencing *LINC01128* inhibits glioma cells’ proliferation, migration, and invasion levels by targeting miR-27B-3p [102]. miR-181B2 is downregulated in the CSF of AD patients [59]. Memory impairment, Aβ aggregation, tau hyper-phosphorylation, neuroinflammation, dysregulation of insulin signaling, and the downregulation of miR-26a, miR-124, miR-29a, miR-181b, miR-125b, miR-132, and miR-146a are observed in the hippocampus of rats exposed to Aβ oligomers. However, these effects are mitigated by treatment with intranasal insulin [103]. In one study, miR-23B was downregulated in blood and brain samples of AD patients and transgenic mice [94,104]. miR-23b-3p ameliorated cognitive deficits, inhibited cell apoptosis, and reduced tau hyperphosphorylation via the downregulation of GSK-3β signaling pathway [104]. In addition, its overexpression improved learning and memory by attenuating long-term neurological deficits and inhibiting the activation of neuronal autophagy [105]. Both miR-LET7A and miR-LET7F1 exhibited decreased expression levels during the presymptomatic stage. Increased expression of let-7a resulted in dysfunctional autophagy and programmed cell death, reduced cell survival, and intensified neurotoxicity in PC12 and SK-N-SH cells, with increases induced by Aβ1-40 [106]. Furthermore, the expression of let-7a, let-7b, and let-7e were upregulated in the progression of AD in a rabbit model [107]. Also, let-7a-3p was upregulated in the CSF of a Chinese Family with PSEN1 G378E mutation [18]. Let-7f was upregulated in the hippocampus of AD patients [22]. Nevertheless, when bone marrow mesenchymal stem cells (MSCs) were subjected to Aβ25−35 in a laboratory setting, they exhibited noticeable early cell death, accompanied by a reduction in the levels of let-7f-5p and an increase in caspase-3 expression [108]. This study discovered that miR-1306 is a protective microRNA against Alzheimer’s disease. The downregulation of CircAXL reduced the harmful effects of Aβ1-42 on SK-N-SH cells, including cytotoxicity, cell apoptosis, inflammation, oxidative stress, and ER stress. This protective effect was partially achieved by increasing the levels of miR-1306-5p [109]. It has been observed to be downregulated in plasma EVs, serum exosomes from AD patients, and Aβ1-42-treated SK-N-SH cells [109,110]. Consistent with our findings, miR-100 is found to be downregulated in the brains, plasma extracellular vesicles (EVs), and neural-derived small EVs from patients with Alzheimer’s disease (AD) [111,112,113]. The expression of miR-100 is modulated by endoplasmic reticulum stress in the brains of APP/PS1 mice. It is reduced during early stages (6–9 months old) and elevated during late stages (12–15 months old) compared to age-matched WT mice. Furthermore, it suppresses mTOR expression, another crucial molecule in Alzheimer’s disease [114]. 

Among the 30 miRNAs that showed increased expression in the presymptomatic stage, miR-3142, miR-4635, and miR-130B were found to be the most highly overexpressed, with fold changes of 5.41, 5.33, and 4.97, respectively. Below, we will discuss miR-130B, as miRNA shows the highest increase in expression in the group of individuals carrying the PSEN1(A431E) mutation. miR-3142 increases expression levels in symptomatic and asymptomatic stages A2 and A3 if we consider a fold change (FC) of at least 1.5. The long non-coding RNA *RMRP* acts as a sponge for miR-3142 and is increased in Alzheimer’s disease. It enhances the expression of *TRIB3*. Knocking down *RMRP* inhibits autophagy and apoptosis through the miR-3142/TRIB3 axis in SH-SY5Y cells treated with Aβ1-42 [115]. CircTAB2, a sponge for miR-3142, is observed to be downregulated in both lung cancer tissue and cells. This phenomenon suggests that miRNA3142 is likely to be overexpressed, as previously reported in the serum of cervical cancer patients [116]. This overexpression aims to reduce the inhibitory impact of miR-3142 on *GLIS* family zinc finger 2 (GLIS2), which in turn prevents the activation of the AKT pathway and thus hinders the progression of lung cancer [117]. No literature about the association between AD and Mir-4635 was found. However, it was observed that this miRNA was upregulated in the lung carcinoma patients’ tissue [118]. miR-425, miR-7-2, and miR-218-2 are other interesting, downregulated miRNAs already reported in AD. miR-425 is decreased in the brain and hippocampus of AD patients and APP/PS1 mice [64,96,119]. An miR-425-deficient mouse model enhanced APP amyloidogenic processing and increased reactive gliosis, thus leading to neuroinflammation, cognitive impairment, and neuron loss by suppressing the PI3K-Akt signaling pathway [119]. Conversely, the overexpression of miR-425-5p significantly increased cell apoptosis, stimulated glycogen synthase kinase-3β (GSK-3β), and elevated tau phosphorylation in HEK293/tau cells [120]. The expression of miR-7 is increased in the brains of patients with Alzheimer’s disease (AD) and with resistance to insulin, as it targets crucial regulators of insulin homeostasis [121] and also key cholesterol biosynthesis elements essential for forming lipid rafts [122]. miR-218 accumulates in the hippocampus and is activated during neuronal differentiation [123,124]. Moreover, the repression of miR-218 increases the quantity and functionality of various receptor tyrosine kinase effectors [125].

### 3.3. miRNAs Altered in MSCs Derived from the PSEN1(A431E) Asymptomatic Mutation Carriers (A2 and A3)

Our findings indicate that asymptomatic conditions have more altered miRNAs, with 81 and 97 for the A3 and A2 stages, respectively. Furthermore, over 50% of the miRNAs exhibit upregulation compared to the other stages. Specifically, 44 miRNAs were identified as overexpressed in the A3 stage, while 58 were found to be overexpressed in the A2 condition. Despite the limited information available regarding the highly modified miRNAs in these stages of Alzheimer’s disease (AD), it is crucial to consider them as potential early indicators of AD in the future. In the A3 stage, miR-4660 (fold change = 6.39) and miR-4293 (fold change = 4.06) exhibited the highest fold changes. In the A2 stage, miR-543 (fold change = 7.16), miR-376B (fold change = 5.73), and miR-154 (fold change = −4.05) displayed the highest fold changes. The miRNAs that showed the most significant decrease in expression in stage A3 were miR-3189 (fold change = −5.48), miR-4799 (fold change = −4.8), and miR-1234 (fold change = −4.65). In stage A2, the miRNAs miR-143 and miR-145 had a fold change of −4.05. The sole microRNA (miRNA) that has been linked to Alzheimer’s disease (AD) is miR-376b-3p, which showed an increase in expression in the urinary exosomes of 5XFAD mice [126]. Nevertheless, several of these miRNAs experience abnormal regulation in cancer and primarily control the growth and spread of tumor cells. Specific examples include miR-143 and miR-145, which will be further examined in the following discussion. The overexpression of miR-4660 in triple-negative breast cancer cells suppressed cell proliferation by targeting the mammalian rapamycin target (mTOR) [127]. Meanwhile, miR-4293 expression is enhanced in lung carcinoma tissue and cells, promoting cell proliferation and metastasis but suppressing apoptosis [128].

Among the modified miRNAs during the asymptomatic stage, certain ones have received extensive research attention in relation to Alzheimer’s disease (AD), including miR-132. This particular miRNA is regarded as a protective factor in AD, as its increased expression helps to alleviate memory impairments, promote neurogenesis, and reduce amyloid accumulation and cell death [129,130,131]. It has been reported as being downregulated in brain and plasma exosomes of AD patients [132]. Similarly, miR-708 is decreased in CSF from patients with AD and in peripheral blood mononuclear cells from FAD patients [133,134]. Increased expression of miR-708 following suppression of *MINCR* (MYC-induced long non-coding RNA) leads to a decrease in CTNNB1 levels, inhibiting the Wnt/β-catenin signaling pathway [135]. The overexpression of these miRNAs in A3 asymptomatic carriers may indicate the presence of a protective or compensatory mechanism. Our findings confirmed that miR-222 was consistently downregulated in APPswe/PSΔE9 mice. This decrease in miR-222 prevented the progression of the cell cycle from the G1 to S phase by binding to CDK2 and cyclin E complexes [136]. In a mouse model induced by Aβ1-42, the microRNA miR-25 hindered the growth of neural cells, leading to cell death, and worsened the damage to neurons by reducing the expression of *KLF2* through the Nrfs signaling pathway [137]. The level of this specific miRNA is diminished in asymptomatic carrier A3, indicating a potential promotion of cell proliferation.

On the other hand, miR-9-1 is highly expressed in the youngest asymptomatic carrier, A2, and this miRNA is crucial for the proliferation and differentiation of neural stem cells. Overexpression of miR-9 leads to a decrease in Hes1 protein levels, which triggers neuronal differentiation and the termination of the cell cycle [138]. miR-485 is detected in abnormal levels in the cerebrospinal fluid (CSF), plasma, and serum of patients with Alzheimer’s disease (AD). It regulates various processes, such as beta-amyloid (Aβ) production, synaptic function, cognitive and motor behaviors, neuroinflammation, and apoptosis [139]. Its expression gradually decreases with AD progression, so it is considered another protective miRNA [64] also upregulated in the A2 asymptomatic stage. miR-33 promotes Aβ secretion and impairs Aβ clearance in neural cells [140]. Its knockdown suppresses inflammation, oxidative stress, cell apoptosis, and improves synaptic plasticity by suppressing the activation of the Akt/mTOR signaling pathway [141]. miR-29B is downregulated in the brain. Plasmatic EVs [142], exosomes charged with miR-29B, demonstrate potential therapeutic benefits, reducing the pathological effects of Aβ oligomers [143]. The asymptomatic stage may exhibit a protective profile of miRNAs based on the information provided.

### 3.4. miRNAs Altered in PSEN1(A431E) Mutation Carriers Compared to Controls

When we compared the mean expression levels of all individuals carrying the PSEN1(A431E) mutation as a collective group with those of the healthy controls, the miRNA miR-130B exhibited the highest fold change (FC) of 4.1 and was the only one that showed upregulation in all conditions. In contrast to our findings, miR-130b is reduced in the plasma of patients with Alzheimer’s disease [144]. However, the expression of miR-130b is increased in primary hippocampal neurons of senescence-accelerated SAMP8 mice induced with H_2_O_2_ [145]. When a fold-change (FC) value of at least 1.5 is taken into account, it is observed that five miRNAs, namely miR-147B, miR-3142 (which has already been mentioned earlier), miR-4429, miR-146A, and miR-4540, are consistently upregulated across all stages. The last one exhibits a highly intriguing trend, decreasing as the disease advances despite being excessively expressed (Figure 9B). miR-146A is widely studied in the context of Alzheimer’s disease (AD) due to its role as a proinflammatory microRNA [146,147]. Similar to our results, it has been reported upregulated in different zones of the brain [97,148,149], but a study reported its upregulation in the temporal cortex of both mild (Braak III) and severe AD (Braak VI) patients [150] and in plasma EVs [84].

Furthermore, the miRNAs miR-4521, miR-1178, miR-4668, and miR-32 exhibited significant downregulation in all conditions, with a fold change of −1.5 or lower (Figure 9C). The miRNAs miR-4521 and miR-1178 were previously mentioned as being among the most downregulated in the presymptomatic and symptomatic stages. The expression of miR-32-5p was increased in the frontal cortex of patients with Alzheimer’s disease [132]. Also, miR-3614 was reported to be upregulated in peripheral blood mononuclear cells (PBMC) of FAD patients [134], and miR-3614, along with miR-1244, was also downregulated in all the conditions.

miR-143, miR-145, miR-4438, and miR-4677 exhibited a collective decrease in expression. However, it is noteworthy that they were downregulated in all conditions except in the symptomatic carrier. This observation is intriguing because inhibiting miR-143-3p promotes the survival of neurons in a cell model in vitro by targeting NRG1 [151], which is overexpressed in plasma EVs and PBMCs from patients with AD [110,134]. Furthermore, miR-145 exhibits increased expression in brain samples affected by Alzheimer’s disease [96]. On the contrary, miR-4635, miR-548AB, and miR-4518 exhibit increased expression in all conditions except the symptomatic carrier.

### 3.5. PSEN1(A431E) Mutation Carriers Enriched Signaling Pathways and Deregulated miRNAs’ Target Genes

In all four conditions, *WEE1*, *BTG2*, *DICER1*, and *ANKRD52* experienced increased regulatory pressure. Wee1, a mitotic regulator, was found to be upregulated in the brains of patients with Alzheimer’s disease. Additionally, it exhibited decreased activity, hyperphosphorylation, and redistribution to the cytoplasm of neurons [152,153]. *BTG2*, also known as B-cell translocation gene 2 or PC3/Tis21, is a protein that inhibits cell proliferation and is controlled by the activation of p53 [154]. It also regulates cell-cycle progression, apoptosis, and differentiation and is involved in the induction of cellular senescence and the response against oxidative stress [155,156]. *DICER1* is a microRNA-processing enzyme that plays critical roles in neuronal survival and neuritogenesis. It is associated with oxidative stress and is decreased in the hippocampi of the AD mouse model APPswe/PSEN1dE9 [157]. *ANKRD52* is a tumor suppressor that regulates PP6c-mediated PAK1 dephosphorylation, which inhibits proliferation [158]. The mRNA alterations in the mutation carriers result in the deregulation of the targets *PTEN*, *CDK6*, *CDKN1A*, and *CCND1*, all subjected to increased regulatory pressure in all four conditions. The carrier that showed symptoms increased the expression of *PTEN*, *CDK6*, and *CDKN1A* genes, with fold changes (FCs) of 2.01, 4.06, and 3.03, respectively. The presymptomatic carrier showed decreased *CDK6* with a fold change (FC) of −2.58, while *CDKN1A* and *CCND1* demonstrated overexpression with FCs of 3.22 and 2.58, respectively. *PTEN* is an additional tumor suppressor and a down regulator of the PI3K/AKT signaling pathway. Akt activity is increased in Alzheimer’s disease. PTEN is an enzyme that facilitates the removal of phosphate groups from PIP3, converting it into PIP2. This enzymatic activity is crucial for regulating cell growth, division, survival, and differentiation [159], and can be activated by multiple kinases, including CK2, GSK3β, and ROCK2 kinase [160,161]. CDK6 is a serine/threonine protein kinase essential for cell cycle G1 phase progression and G1/S transition. YAP-CDK6 signaling is downregulated in aged and AD model mice, promoting cellular senescence [162]. CDKN1A is a cyclin-dependent kinase inhibitor that regulates cell cycle progression at G1 and plays a regulatory role in S-phase DNA replication and damage repair. CDKN1A is upregulated by Aβ plaque-associated oligodendrocyte progenitor cells that exhibit a senescence-like phenotype in the brains of patients with AD and those of AD mouse models [163]. CCND1 is a cyclin that forms a complex with CDK6 or CDK4, whose activity is required for cell cycle G1/S transition and localized in neurons of the AD brain [164,165]. Many genes face high regulatory pressure, particularly influencing cell cycle processes, cell growth, and proliferation, which aligns with the predominantly enriched pathways: cellular senescence, p53, and cell cycle signaling pathways. Furthermore, the FoxO and NF/kappa B signaling pathways are also implicated in neuroinflammation and proliferation. The Toll-like receptor and PI3K-Akt signaling pathways are involved in regulating apoptosis.

## 4. Materials and Methods

### 4.1. Subjects of Study

A study was conducted on a Mexican-mestizo family with a three-generation history of paraparesis spastic. Four of the eight subjects studied had the PSEN1(A431E) mutation, while the other four were age-matched individuals without dementia who served as controls. The samples used in this study consisted of IDs from two groups: patients (P) and controls (C). The patients’ ages when isolating MSCs were P18, P36, P44, and P54, while the corresponding control ages were C18, C35, C42, and C55. The carriers and patients were categorized as follows: asymptomatic (P15 and P36), presymptomatic (P44) with only hyperreflexia and slight hyposmia, and symptomatic (P54) with spastic paraparesis, anosmia, and slight cognitive impairment.

The study protocol received approval from the Bioethics Committee on Human Research (COBISH, folio #038/2016) of the CINVESTAV-IPN. Before the genetic investigation, written informed consent was obtained from all subjects. Furthermore, our group previously conducted a cognitive and neurological assessment and a genetic confirmation of the A431E mutation in the *PSEN1* gene [34].

### 4.2. Sample Collection and RNA Extraction

The MSCs were obtained and cultured following the protocol previously described [24]. They were harvested in Dulbecco’s modified Eagle and F-12 media (DMEM/F-12 Gibco, Grand Island, NY, USA), which were supplemented with 10% fetal bovine serum (Gibco, Grand Island, NY, USA), 4 mM GlutaMAX (Gibco, Paisley, UK), 100 g/mL Streptomycin, and 100 IU/mL Penicillin (SIGMA, Saint Louis, MI, USA). The cells were maintained at 37 °C in a humidified atmosphere with 5% CO_2_. Before acquiring the sample, we subjected the dishes to a two-hour treatment with poly-D-lysine diluted in Milli-Q water at a concentration of 50 mg/mL. Upon reaching 80% confluence, the cells were transferred to a new plate and cultured under identical conditions.

Per the manufacturer’s guidelines, we isolated total RNA using TRIzol reagent BD (Invitrogen Corp., Carlsbad, CA, USA) and purified it using an RNeasy kit (Qiagen Inc., Valencia CA, USA). The quantification of RNA was performed using a Nanodrop-ND-2000c instrument (Thermo Scientific, Wilmington, DE, USA), while the evaluation of RNA integrity was conducted using a Bioanalyzer 2100 instrument with the RNA 6000 Pico kit (Agilent, Santa Clara, CA, USA). Only samples that fell within the specified ranges for purity and integrity were included. These ranges were defined as having an A260/A280 ratio between 1.8 and 2.1, an A260/A230 ratio of 2 to 2.2, and an RIN value of 8 or higher.

### 4.3. Microarray Assays

The Affymetrix platform was employed to examine the expression of miRNAs and their target genes using the GeneChip Human Gene 2.0 ST microarray(Affymetrix, Santa Clara, CA, USA) and the whole transcript (WT) plus reagent kit. The array comprises around 48,000 probesets, enabling precise, sensitive, and extensive assessment of protein-coding and long intergenic non-coding RNA transcripts. The dataset includes approximately 25,000 mRNA molecules, which accounts for about 83% of the total reported in the Human Genome Project (HGP). About 11,000 lncRNA molecules represent around 73% of the total. The dataset also contains 313 snoRNAs, accounting for 57% of the total, 42 snRNAs, representing 21%, and 1250 pre-miRNAs, which comprise approximately 65%.

The analysis was conducted on 100 ng of total RNA following the manufacturer’s instructions. Figure 12 the sequential steps in the samples’ amplification, labeling, fragmentation, and hybridization process in the microarrays. These microarrays employ probes consisting of 25 nucleotides of single-stranded DNA. These probes selectively bind to specific gene regions. By utilizing the principle of hybridization and complementarity between the probe and the complementary DNA, which was synthesized from the target RNA, we could determine the expression of transcripts by examining their hybridization with different regions of the DNA. The expression level for each transcript was directly proportional to the fluorescence emitted during the hybridization of the probes with the target transcript sequence. It was determined through image analysis and represented as an average intensity value (log2) of the probeset.

### 4.4. Data Analysis

The datasets were analyzed using CEL files and the Transcriptome Analysis Console (TAC) v. 4.0 software from Affymetrix, located in Santa Clara, CA, USA. The probe sets were summarized using the median polish method and normalized using quantiles. The background noise correction was accomplished using the robust multi-chip average (RMA) method, and the data were subsequently transformed using a logarithmic base 2 transformation. The identification of differentially expressed genes was performed using ANOVA. Genes were classified as altered if their fold change was equal to or greater than 2, or equal to or less than −2, and if their *p*-value was less than 0.05, comparing the patient group to the control group. If the transcripts had more than one probeset to evaluate their expression and the alteration was in the same direction, those transcripts mapped with the most significant number of probes were included. If this number was indistinct for said transcripts, those with a fold change with the most considerable magnitude (up or down) were selected. Uncharacterized probesets (LOC), for which any functional annotation is unknown or not described, pseudogenes, as well those transcripts from different probesets whose fold change was controversial (expressed upwards in one of them and expressed downwards in the other) were removed (Table 1 and Appendix A).

We combined all the transcripts from the stage disease data to analyze them as a group. We calculated the average differential expression of PSEN1(A431E) mutation carriers compared to the average of MSCs without the mutation. We then selected the transcripts with a fold-change (FC) value greater than or equal to 1.5 or less than or equal to −1.5.

### 4.5. Prediction and Validation of Target mRNAs

In order to determine the targets of the specific miRNAs affected by the β(A431E) mutation in carriers compared to controls, we searched for the predicted targets of the miRNAs in each group using five different databases. We utilized TargetScan human v.8.0, miRDB v. 6.0, and DIANA-microT v. 2023 to identify putative genes. Additionally, we utilized TarBase v.8.0 and miRTarBase v.9.0 to obtain data on targets undergoing experimental validation and verification through techniques such as qPCR, Western blot, enzyme-linked immunosorbent assays, and microarrays. This was necessary because not all predicted gene results could be considered reliable. Due to the excessive amount of data obtained, the Excel program’s capacity was exceeded. A filtering algorithm was incorporated into the MATLAB program (Appendix A), leading to refined lists of miRNA targets for each group. Subsequently, Venn diagrams were utilized to compare the results from the three databases for each group. The intersection of the diagrams was then selected to identify the putative genes for the following bioinformatic analyses. Additional Venn diagrams were utilized to identify the overlap between the putative, validated, and altered targets in MSCs with the PSEN1(A431E) mutations (Appendix A).

### 4.6. Bioinformatic Analysis

The WEB-based Gene SeT AnaLysis Toolkit (WEBGESTALT v. 2019) (http://www.webgestalt.org/, accessed on 18 december 2023) was used to conduct functional and biological analysis, in which the DE miRNAs and mRNAs were involved with the Kyoto Encyclopedia of Genes and Genomes (KEGG v.108.1 accessed on 18 december 2023), REACTOME v.86 https://reactome.org/, accessed on 18 december 2023, PANTHER v.18.0 (http://pantherdb.org/, accessed on 18 december 2023) and Wikipathways v. 2023 databases accessed on 18 december 2023. This was performed individually and by group using FDR significance level. DAVID (Database for Annotation, Visualization, and Integrated Discovery) for the diseases with GAD database.

All the images were made with biorender.com.

## 5. Conclusions

miRNAs play a crucial role in the development of neurodegenerative diseases and can serve as biomarkers for disease diagnosis and potential targets for therapy, as well as aid in understanding the underlying molecular and cellular processes driving disease progression.

This study presents precise miRNA expression profiles in cells derived from patients with familial Alzheimer’s disease (FAD) across various stages of symptoms. Our analysis enables us to differentiate between the carrier condition of the A431E mutation and the control condition (individuals without the mutation). It provides a comprehensive overview of the relevant miRNAs that have already been reported in AD. This comprehension can illuminate the fundamental mechanisms that contribute to the progression of the illness. Furthermore, these findings confirm that olfactory mesenchymal stem cells (MSCs) have the potential to serve as a valuable model for studying Alzheimer’s disease (AD). However, additional investigations are required to develop innovative therapeutic approaches and biomarkers.

## 6. Patents

(1) Mx/a/2021/014065; (2) Mx/a/2021/014066; (3) Mx/a/2021/014067; and (4) Mx/a/2021/014068.

## Figures and Tables

**Figure 1 ijms-25-01580-f001:**
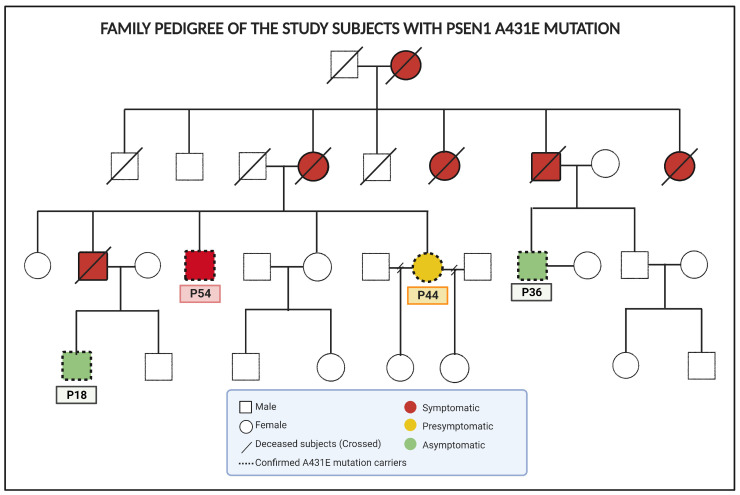
Family pedigree of the study subjects with PSEN1(A431E) mutation. Modified from [34].

**Figure 2 ijms-25-01580-f002:**
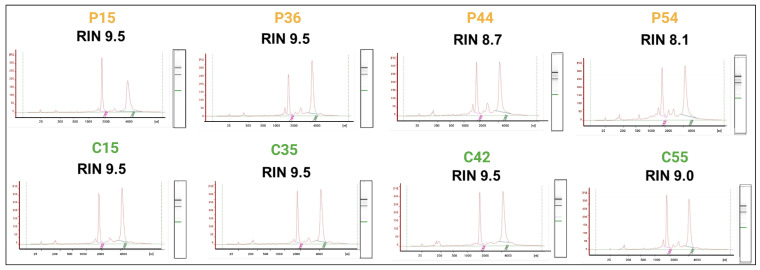
Evaluation of the integrity of the total RNA samples. Yellow letters correspond to the PSEN1(A431E) mutation carriers (up) and green letters correspond to the healthy controls. The electropherograms display the various species of RNAs present in the total RNA extractions, ranging from ribosomal RNAs to small RNAs less than 200 nucleotides in length. Additionally, the gel-type image highlights the prominent ribosomal RNA bands. We did not detect any genomic DNA (gDNA).

**Figure 3 ijms-25-01580-f003:**
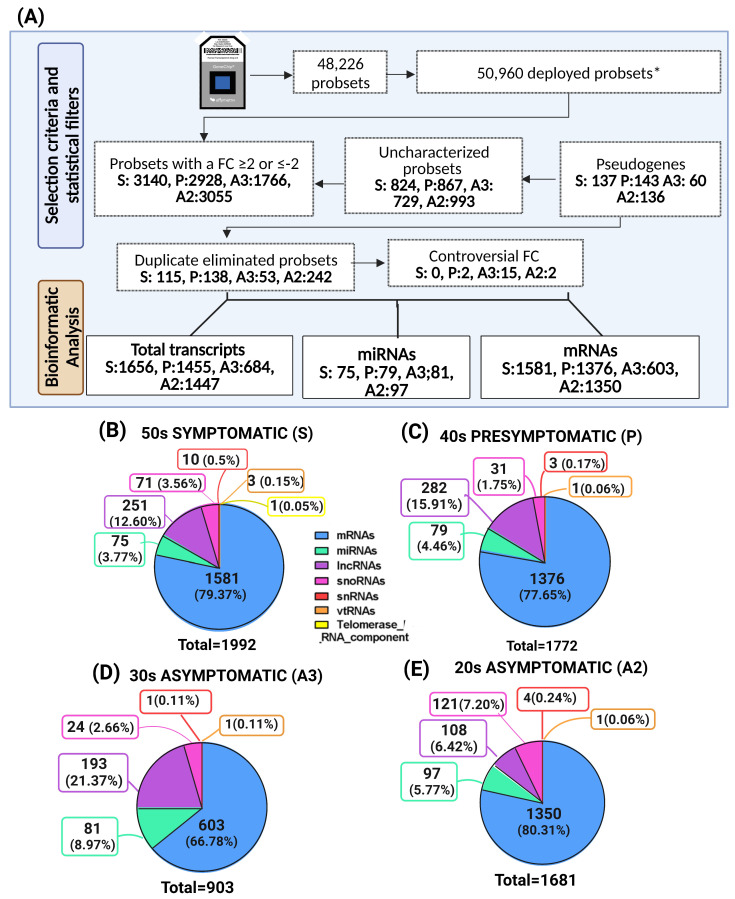
Selection and types of transcripts by disease stage. (**A**) The selection criteria were employed using the data obtained from the microarrays to determine the number of transcripts that would be examined. (**B**–**E**) display the categorization of transcripts and the corresponding distribution percentages according to groups. S represents the symptomatic stage, while P represents the presymptomatic stage. A3 refers to the asymptomatic stage in individuals who are 30 years old, while A2 refers to the asymptomatic stage in individuals who are 20 years old. lncRNA stands for long non-coding RNA, snoRNA stands for small nucleolar RNA, snRNA stands for small nuclear RNA, and vtRNA refers to vault RNA. * Deployed probesets were obtained by searching functional annotations of uncharacterized probesets in the DAVID, NCBI, and Biotools databases to access more results.

**Figure 4 ijms-25-01580-f004:**
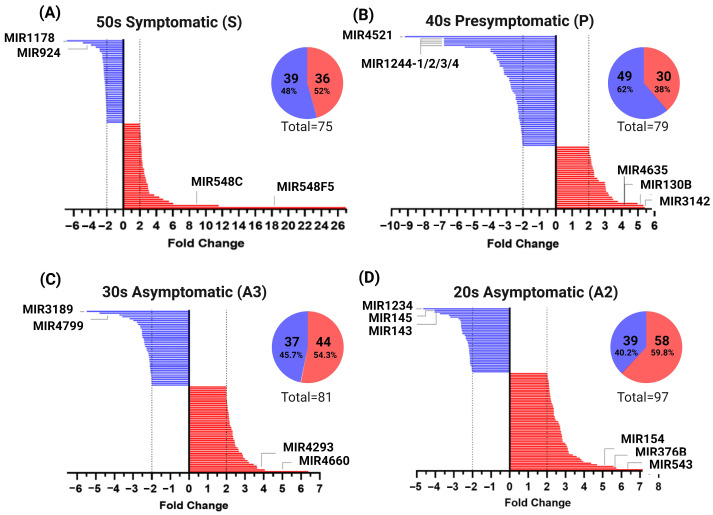
miRNAs expression profiles by the groups. The miRNAs with upregulated expression are depicted in red, while those with downregulated expression are in blue. (**A**) Individuals exhibiting symptoms. (**B**) Individuals in their 40s who have not yet developed disease symptoms may do so soon. (**C**) Individuals in their 30s who do not show any symptoms of the disease. (**D**) Individuals in their 20s who do not show any symptoms of the disease.

**Figure 5 ijms-25-01580-f005:**
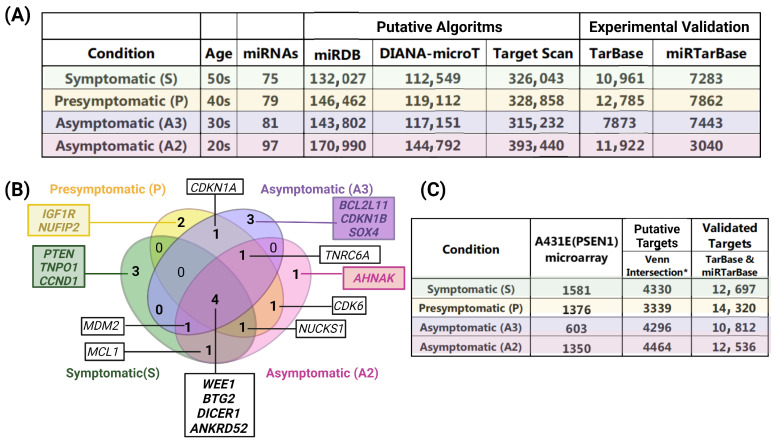
Putative and validated target interactions and genes for the altered miRNAs by group. (**A**) The table displays the putative and validated target interactions categorized by the group in each database utilized. The putative interactions were obtained from mirDB, DIANA-microT, and Target Scan, while the validated interactions were sourced from TarBase and miRTarBase. (**B**) Venn diagram illustrating the top 10 genes that experience higher regulatory pressure from miRNAs. In green the symptomatic stage, in yellow the presymptomatic stage and in purple and pink the asymptomatic A3 and A2 stages respectively (**C**) Table displaying the distinct target genes that were putative and validated for each stage. * Number of targets shared between mirDB, DIANA-microT, and Target Scan databases.

**Figure 6 ijms-25-01580-f006:**
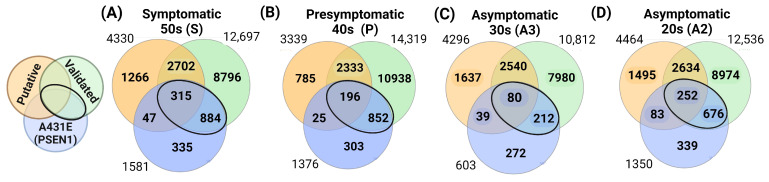
Putative and validated target genes that coincide with the altered transcripts in MSCs of PSEN1(A431E) mutation carriers. (**A**) Symptomatic S, (**B**) Presymptomatic P, (**C**) Asymptomatic A3, (**D**) Asymptomatic A2.

**Figure 7 ijms-25-01580-f007:**
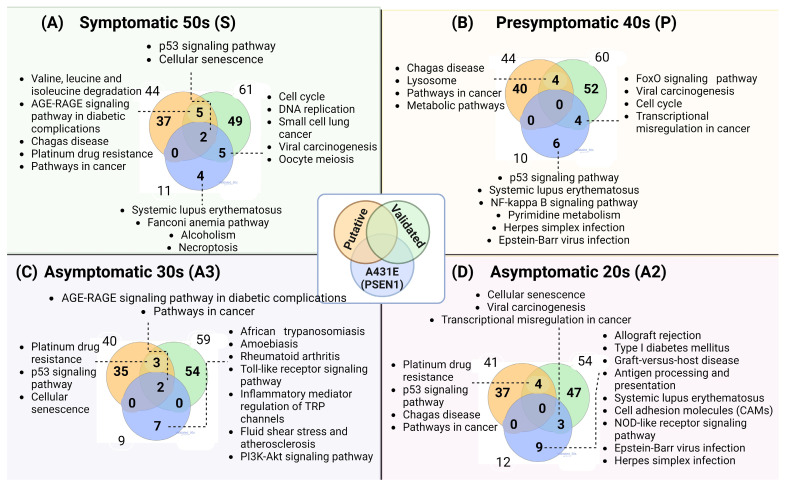
Enriched pathways in putative, validated, and altered mRNAs in PSEN1(A431E) mutation carriers by stage. (**A**) 50s Symptomatic S, (**B**) 40s Presymptomatic P, (**C**) 30s Asymptomatic A3, (**D**) 20s Asymptomatic A2. Outside numbers indicate the pathways resulted with the targets in each database and the numbers inside the shared pathways between databases.

**Figure 8 ijms-25-01580-f008:**
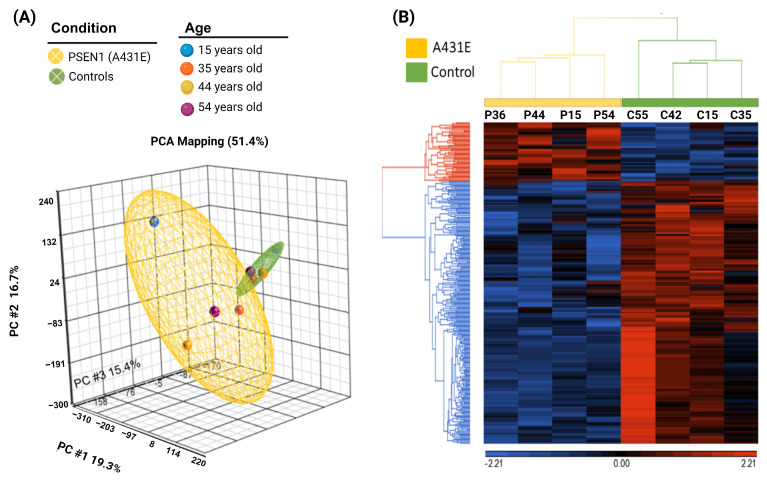
Transcriptomic profile of MSCs derived from PSEN1(A431E) mutation carriers and controls. (**A**) Principal component analysis (PCA) of miRNA expression profiles in MSCs carrying the PSEN1(A431E) mutation. Each point represents a sample, and colors correspond to age groups. The yellow ellipsoid represents cells from individuals with the PSEN1(A431E) mutation, and the green one represents cells from control individuals. (**B**) Heat map of the expression profile of coding and non-coding RNAs differentially expressed in MSCs from healthy individuals and individuals affected by the A431E mutation. Each gene is represented in rows, and each sample is displayed in columns. Average linkage calculated the distance between the two groups. Genes without differences in expression have zero value and are represented in black. Genes with increased expression are depicted in red, while those with reduced expression are shown in blue. The data were standardized so that the average of each transcript was zero and the standard deviation was 1, ensuring all transcripts had the same weight. Graphs were generated with Partek v6.6beta software accessed on 9 April 2019).

**Figure 9 ijms-25-01580-f009:**
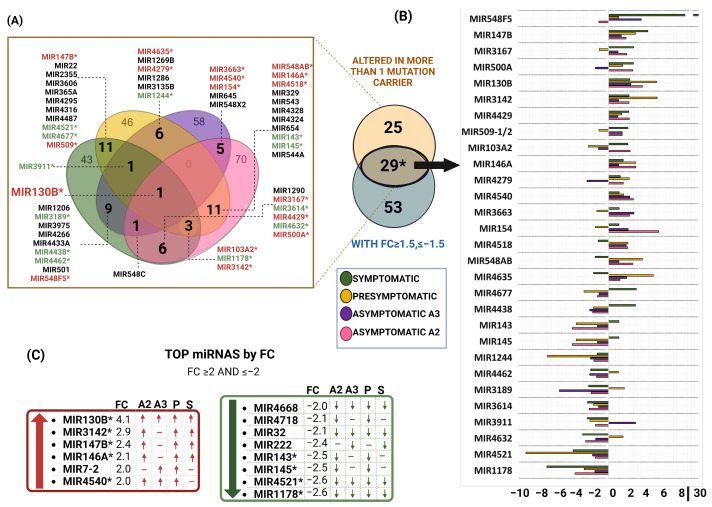
miRNA profile of PSEN1(A431E) mutation carriers. (**A**) Venn diagram of miRNAs altered by symptomatologic stage. In green the symptomatic stage, in yellow the presymptomatic stage and in purple and pink the asymptomatic A3 and A2 stages respectively. MiRNAs shared in more than 1 mutation carrier are in bold and correspond to the 54 miRNAs in the orange circle of Venn diagram in figure (**B**), blue circle shows miRNAs with a FC ≥ 1.5 or ≤−1.5. This FC was obtained with the average differential expression of PSEN1(A431E) mutation carriers versus the average of MSCs without the mutation. The 29 resulting miRNAs are shown in the graphic, where green corresponds to the symptomatic stage, yellow to presymptomatic, and purple and pink to the asymptomatic stages A3 and A2, respectively. (**C**) Top miRNAs by FC (FC ≥ 2, ≤−2). * Correspond to the 29 shared miRNAs, and miRNAs with red letters are upregulated and green downregulated.

**Figure 10 ijms-25-01580-f010:**
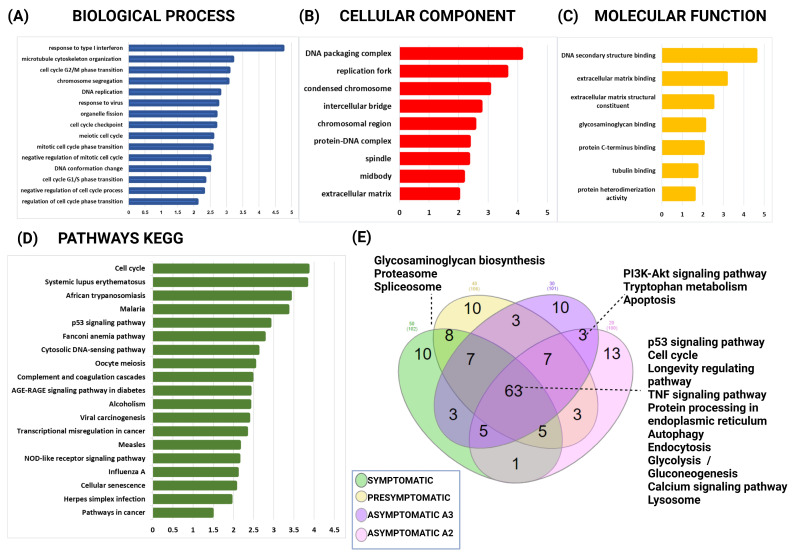
Gene ontology annotation and signaling pathways enriched with the targets of miRNAs altered in MSCs from PSEN1(A431E) mutation carriers. (**A**) Biological process (BP), (**B**) cellular component (CC), (**C**) molecular function (MF), (**D**) KEGG-enriched signaling pathways (KESP), (**E**) shared signaling pathways between symptomatologic stages.

**Figure 11 ijms-25-01580-f011:**
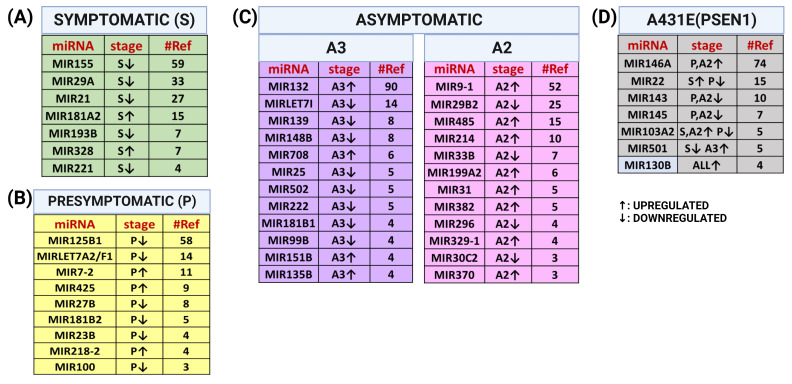
miRNAs altered in MSCs from PSEN1(A431E) mutation carriers linked to AD previously reported. Tables with the miRNAs previously reported are linked to the AD number of references found in the PUBMED search. (**A**) Symptomatic S (green), (**B**) Presymptomatic P (yellow), (**C**) Asymptomatic stages A3 (purple), and A2 (pink). (**D**) miRNAs changed in more than 1 disease condition. ↑: miRNA upregulated and ↓: downregulated. Only the miRNAs with more than four references are shown.

**Figure 12 ijms-25-01580-f012:**
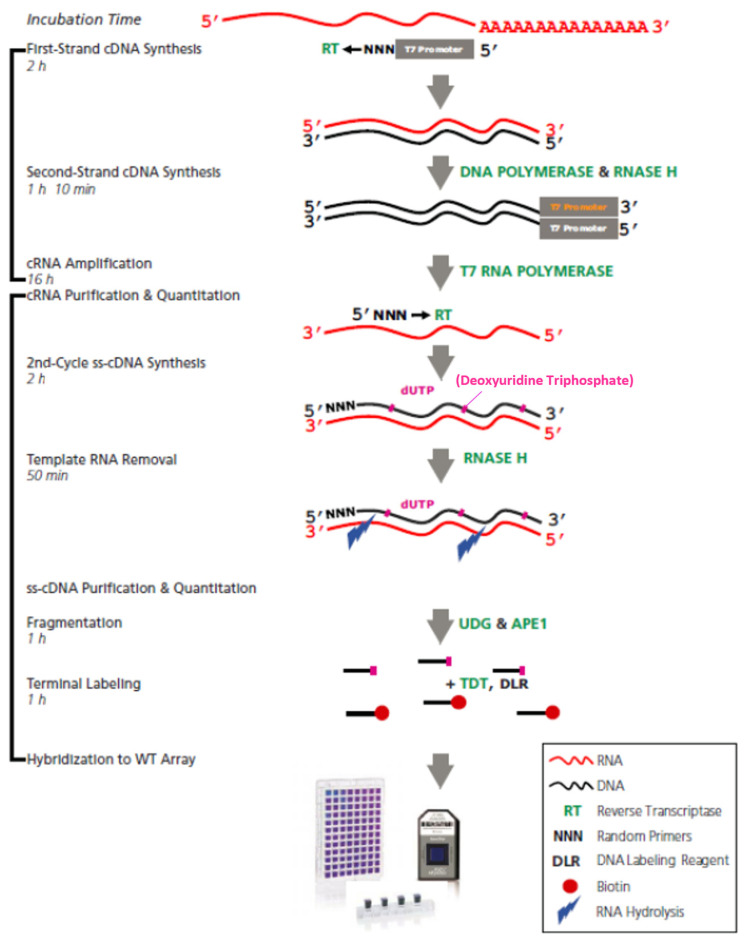
Diagram showing the steps involved in labeling, amplifying, fragmentation, labelling and hybridizing the microarrays.

**Table 1 ijms-25-01580-t001:** Transcripts that were removed by stage of disease.

Filtered by FC (≤2, ≥−2)	3140	2928	1766	3055
	Symptomatic S	Presymptomatic P	Asymptomatic A3	Asymptomatic A2
Repeated	115	138	53	242
Controversial FC	0	2	15	2
Uncharacterized	824	867	729	993
Pseudogenes	137	143	60	136

## Data Availability

Data are contained within the article and Appendix A.

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
