# Peer review of "Mesenchymal Stem Cells from Familial Alzheimer’s Patients Express MicroRNA Differently"

_ijms, 2024, doi:10.3390/ijms25031580_

Round 1
Reviewer 1 Report
Comments and Suggestions for Authors
In this study, the authors employed microarrays to analyze the transcriptome of nasal epithelial cells of the participants carrying the PSEN1 (A431E) mutation that causes familial form of Alzheimer’s disease. They identified the mRNAs and miRNAs with altered levels, determined their target genes at asymptomatic, pre-symptomatic, and symptomatic stages and then further compared with those from age-matched healthy controls. With the data, they also performed diverse bioinformatic analyses to assess their biological processes and pathways of the differentially expressed genes. Despite being of interesting, the work has some major concerns as listed below.
Major concerns:
1. From the title through to the entire manuscript, authors indicated that mesenchymal stem cells (MSCs) were used in their study; however, it appears that they never sorted the nasal cells by FACS and only used the mixed cell population, i.e. nasal epithelial cells. Is there any evidence that the mixed nasal epithelial cells are basically MSCs? Based on their Materials and Methods, it seems that they didn’t perform cell soring assay to purify MSCs to use the purified MSCs for this study. Accordingly, use of the term MSCs in this manuscript is inappropriate.
2. Introduction is too long.
3. Lines 209-213 should include Figure information in order to support their results.
4. Results shown in Fig 4D were opposite to the description included in Lines 224-227.
5. The description included in Lines 288-291 is not consistent with the data shown in Figures 6.
6. Authors identified numerous up- and down-regulated miRNAs at different stages of AD but they never validated any of the identified results.
7. As the results were obtained from mixed population of nasal epithelial cells, it would be more meaningful if the results were obtained from the neurons differentiated from the purified nasal MSCs.
8. The last figure is absence of Figure Legend.
Minor concerns:
1. The MS contains multiple typos, such as line 87 (“plasm. [13].”); the similar typos were also seen in lines 92, 102, 108, 114, 119, 123, 149, 473, 491 …
2. Lines 915-919 are duplications of lines 910-914.
Comments on the Quality of English LanguageThere are a number of typos. Somewhere of the manuscript is too long and needs to be concise.
Author Response
Response to Reviewer 1 Comments
Dear Reviewer, We genuinely appreciate the time and effort you dedicated to thoroughly evaluating our manuscript. We are pleased to inform you that we have carefully considered and implemented all of your suggestions into the revised version of the manuscript to improve and refine the quality of our work.
- From the title through to the entire manuscript, authors indicated that mesenchymal stem cells (MSCs) were used in their study; however, it appears that they never sorted the nasal cells by FACS and only used the mixed cell population, i.e. nasal epithelial cells. Is there any evidence that the mixed nasal epithelial cells are basically MSCs? Based on their Materials and Methods, it seems that they didn’t perform cell sorting assay to purify MSCs to use the purified MSCs for this study. Accordingly, use of the term MSCs in this manuscript is inappropriate.
We added some information to make the text more straightforward, but first, let me explain in detail:
Without information about these cells, this observation can be considered noteworthy. We are sorry and appreciate the observation. Properties and characterization of these cells, including differentiation, adherence, and clonogenicity, have already been studied by many groups (Veron et al. 2018; Delorme et al. 2010; Benítez-King et al. 2011 Murrell et al. 2005; 2009; Ge et al. 2015). We used the isolation method of Benítez King and Cols (Benítez-King et al. 2011) through a non-invasive procedure with a toothbrush. It is reported that in the medium conditions that we use, from passage six, these cells show a homogeneous population of MSCs which was confirmed in our recently published paper, where we used the same cells from mutation carriers and controls and characterized them by Flow cytometry to confirm the expression of specific markers for mesenchymal stem cells (CD105, CD90, and CD73) and the absence of hematological markers and differentiated cells (CD34, CD45, CD14, CD19, and CD166) at nine passage. Purifying the cells using FACS is not required because the medium conditions naturally select the cells, resulting in approximately 98-99% of positive cells for MSC markers in passage 6 in all the samples (see our previous report: Rochín-Hernández et al., 2023).
The text was improved as follows:
A limited amount of research has assessed the deregulation of miRNAs in neural precursor cells. A potential source of neural stem cells is the olfactory mucosa, where neurogenesis is necessary to replace the olfactory neurons. Ecto-mesenchymal stem cells (MSCs) show great potential because they can be obtained from the olfactory nasal niche without invasive procedures[24]. According to Benítez King's group, the cells in passage six exhibit a uniform population of MSCs under the specific medium condition.
Furthermore, multiple research groups have extensively investigated the properties and characteristics of these cells, such as differentiation, adherence, and clonogenicity [24–26]. These cells exhibit a higher neurogenic potential than mesodermal lineage due to their ectodermic origin, making them valuable for treating neurodegenerative disorders. This particularity is significant because many patients with Alzheimer's experience a reduced sense of smell years before symptoms appear[24, 27, 28]. Moreover, Aβ and tau protein aggregates have been detected through all the olfactory pathways, olfactory bulb, and neuroepithelium[29–32]. Studies have also shown an altered neurogenesis with lower viability of olfactory neurons compared to controls[33]. Our group recently published a study on the modified proteostasis network and characterized MSCs obtained from individuals carrying the PSEN1 (A431E) mutation before and after developing symptoms. We used two label-free proteomic methods to analyze the data[34].
- Introduction is too long.
We have already implemented your suggestion and reduced the introduction's length.
- Lines 209-213 should include Figure information in order to support their results.
We already included the Figure information.
- Results shown in Fig 4D were opposite to the description included in Lines 224-227.
Yes, thanks for your observations. We made a mistake when editing the miRNAs in the graph, but the description of the data in the text is correct, which can be observed in the supplementary data.
- The description included in Lines 288-291 is not consistent with the data shown in Figure 6.
We forgot to add the description; it corresponds to the intersection highlighted, but we also modified the figure to make it more straightforward.
- Authors identified numerous up- and down-regulated miRNAs at different stages of AD but never validated any of the results they identified.
Dear reviewer, you are correct. We are in the process of continuing this work and validating several of the altered RNAs. However, this will take some time and will require a significant effort. We have validated a few miRNAs, but these are preliminary data, and we haven’t added them to the work. In lines 202-204, we wrote, "Thus, our results should be validated. However, our statistical criteria are more stringent than other transcriptomic studies in AD.”
- As the results were obtained from a mixed population of nasal epithelial cells, obtaining them from the neurons differentiated from the purified nasal MSCs would be more meaningful.
As described in comment 1, we do not have a mixed population and no epithelial cells. One of the essential points of our work is that undifferentiated cells, neural precursors from PSEN1(A431E) mutation carriers, exhibit an altered miRNome compared with cells from healthy donors, which could suggest and help to explain why the neurogenesis is changed and why the patients with FAD have an earlier onset of symptoms. But of course, it is planned to be done since working with neurons derived from these MSCs will be very interesting.
- The last figure is absence of Figure Legend.
The reason why is that it is the graphical abstract
Minor concerns: Thanks for your observations. We have already corrected them.
- The MS contains multiple typos, such as line 87 (“plasm. [13].”); the similar typos were also seen in lines 92, 102, 108, 114, 119, 123, 149, 473, 491 …
- Lines 915-919 are duplications of lines 910-914.
There are a number of typos. Somewhere of the manuscript is too long and needs to be concise.

Reviewer 2 Report
Comments and Suggestions for Authors
The authors present an analysis of the miRNA landscape in mesenchymal stem cells derived from individuals that are carriers of a pathogenic mutation on presenilin-1 compared with wild type individuals. The study has the merit of considering the molecular alterations occurring over age by including both younger and older individuals in their analysis. This is of particular importance due to the multi-decade evolution of neurodegenerative diseases and the scarcity of our knowledge on what occurs in the earlier phases of the disease. One limitation of the study is the small number of samples analyzed.
Overall, the study contributes important new information on the onset and evolution of neurodegenerative diseases. I have only a few minor concerns, listed below, for the authors to address before publication:
1) I recommend that the authors provide additional background information on mesenchymal stem cells in the introduction (lines 118-123). Are these cells part of the peripheral or central nervous system? What do we know about their links with neurodegeneration, are they know to be affected in Alzheimer's disease? Are there previous studies that looked into them and what did they find?
2) The analysis shown in Figure 3B-E (correct last letter from (D) to (E), as letter (D) is repeated twice) shows the numbers of transcripts that are differentially expressed across patients categories based on age and symptoms. The number of DETs are comparable across categories, ranging between 1300 and 1500, except for the asymptomatic individuals in their 30s with 603 DETs. I recommend adding some language to acknowledge that since a small number of samples was analyzed it is difficult to draw any conclusion on a potential correlation between number of DETs and disease stage/mutational status.
3) On the same figure panel (3D), are the asymptomatic in their 30s wild type or mutated?
4) The introduction could be improved by including more background and additional references on emerging methods for analyzing and integrating omics data for the study of neurodegeneration (lines 62-65), including the following:
- DeepOmicsAE: Representing Signaling Modules in Alzheimer's Disease with Deep Learning Analysis of Proteomics, Metabolomics, and Clinical Data. Panizza, E. J Vis Exp. 2023 Dec 15:(202).
- Multi-omics Data Integration, Interpretation, and Its Application. Subramanian I, et al. Bioinform Biol Insights. 2020 Jan 31;14:1177932219899051.
Author Response
Response to Reviewer 2 Comments
Dear Reviewer, we sincerely appreciate your comments and the time and effort you dedicated to reviewing our work. We have carefully considered and implemented your suggestions into the revised version of the manuscript.
- I recommend that the authors provide additional background information on mesenchymal stem cells in the introduction (lines 118-123). Are these cells part of the peripheral or central nervous system? What do we know about their links with neurodegeneration, are they know to be affected in Alzheimer's disease? Are there previous studies that looked into them and what did they find?
For this point, we restructured the paragraph and added some information to provide more straightforward sentences. These cells are MSCs from nose neuroepithelium (we took the sample with a non-invasive procedure with a toothbrush). As it says, the characteristic aggregates in AD are found through all the olfactory pathways, including the neuroepithelium, and neurogenesis is altered in that zone in AD patients. Moreover, it is well known that patients with neurodegenerative diseases present hyposmia years before symptoms appear.
“A limited amount of research has assessed the deregulation of miRNAs in neural precursor cells. A potential source of neural stem cells is the olfactory mucosa, where neurogenesis is necessary to replace the olfactory neurons. Ecto-mesenchymal stem cells (MSCs) show great potential because they can be obtained from the olfactory nasal niche without invasive procedures[24]. According to Benítez King's group, the cells in passage six exhibit a uniform population of MSCs under the specific medium condition.
Furthermore, multiple research groups have extensively investigated the properties and characteristics of these cells, such as differentiation, adherence, and clonogenicity [24–26]. These cells exhibit a higher neurogenic potential than mesodermal lineage due to their ectodermic origin, making them valuable for treating neurodegenerative disorders. This particularity is significant because many patients with Alzheimer's experience a reduced sense of smell years before symptoms appear[24, 27, 28]. Moreover, Aβ and tau protein aggregates have been detected through all the olfactory pathways, olfactory bulb, and neuroepithelium[29–32]. Studies have also shown an altered neurogenesis with lower viability of olfactory neurons compared to controls[33]. Our group recently published a study on the modified proteostasis network and characterized MSCs obtained from individuals carrying the PSEN1 (A431E) mutation before and after developing symptoms. We used two label-free proteomic methods to analyze the data[34]. “
2) The analysis shown in Figure 3B-E (correct last letter from (D) to (E), as letter (D) is repeated twice) shows the numbers of transcripts that are differentially expressed across patients categories based on age and symptoms. The number of DETs are comparable across categories, ranging between 1300 and 1500, except for the asymptomatic individuals in their 30s with 603 DETs. I recommend adding some language to acknowledge that since a small number of samples was analyzed it is difficult to draw any conclusion on a potential correlation between number of DETs and disease stage/mutational status.
We have already corrected the Figure 3.
We added the following paragraph:
The transcripts expressed differentially (DETs) varied across patient categories based on age and symptoms. The number of DETs was similar across categories, ranging from 1300 to 1500, except for asymptomatic individuals in their 30s with 603 DETs. Due to the limited number of samples analyzed, it is challenging to establish a definitive correlation between the number of differentially expressed transcripts (DETs) and the stage of the disease or mutational status. Thus, our results should be validated. However, our statistical criteria are more stringent than other transcriptomic studies in AD.
- On the same figure panel (3D), are the asymptomatic in their 30s wild type or mutated?
The four pie diagrams depict the transcripts modified according to the disease stage. Specifically, these transcripts show a fold change (FC) of -2, +2, or greater when comparing the mutation carrier with their respective healthy control.
4) The introduction could be improved by including more background and additional references on emerging methods for analyzing and integrating omics data for the study of neurodegeneration (lines 62-65), including the following:
- DeepOmicsAE: Representing Signaling Modules in Alzheimer's Disease with Deep Learning Analysis of Proteomics, Metabolomics, and Clinical Data. Panizza, E. J Vis Exp. 2023 Dec 15:(202).
- Multi-omics Data Integration, Interpretation, and Its Application. Subramanian I, et al. Bioinform Biol Insights. 2020 Jan 31;14:1177932219899051.
We have already implemented your suggestions:
The advancement of "omic" sciences, employing cutting-edge techniques like Next-Generation-Sequencing, Mass Spectrometry, microarrays, and deep learning methods, enables us to analyze multi-omics data and gain a holistic and thorough comprehension of diseases like Alzheimer's disease (AD)[9, 10]. Current research has highlighted the importance of epigenetics in the progression of Alzheimer's disease (AD). This study focuses on investigating the influence of DNA methylation, chromatin remodeling, histone modifications, and the control of non-coding RNAs in individuals with Alzheimer's disease (AD), as well as in animal and cellular models that are relevant to this condition[11, 12].
Finally, although it wasn’t a listed concern, we want to comment on the following regarding the limitation due to the number of samples analyzed:
Out of all the PSEN1 mutations, only three mutations are prevalent in causing familial Alzheimer's disease (FAD) in Latin America. The first mutation, E208A, has been documented in Antioquia, Colombia, and is estimated to affect around 5,000 individuals (Sepulveda-Falla, Glatzel, and Lopera 2012). The second population comprises eight families from Puerto Rico who carry the G260A mutation (Athan et al. 2001). Lastly, in Mexico, there is a study on the A431E (c.1292C>A, rs63750083) mutation, which has a founder effect in Jalisco. Thus far, less than ten studies have documented this mutation. Most of these mutations are documented in case reports, which offer phenotypic descriptions accompanied by small pedigrees. The A431E mutation in the PSEN1 gene was initially discovered by Rogaeva in 2002. This mutation was found in five patients, but no information about their clinical history was provided. In a subsequent study, Yescas et al. (2006) identified this genetic mutation in nine distinct Mexican families affected by early-onset Alzheimer's disease. The researchers proposed that the mutation likely originated from a shared ancestor. In the same year, Murrell et al. (2006) documented 20 patients belonging to 15 families who exhibited the A431E mutation. Among these patients, 14 were of Mexican mestizo descent and displayed symptoms similar to those reported by Yescas et al. (2006). In 2019, Parker et al. (2019) documented a case of homozygosity characterized by an exceptionally severe phenotype and early onset. In our recently published paper, Rochín-Hernández et al. (2023), we conducted a comprehensive clinical analysis of individuals with the A431E mutation. These individuals were the focus of our research. They were also utilized in our previous work, where we compared the proteome of mesenchymal stem cells (MSCs) derived from the mutation carriers to those obtained from healthy donors.
References:
Athan, Eleni S., Jennifer Williamson, Alejandra Ciappa, Vincent Santana, Stavra N. Romas, Joseph H. Lee, Haydee Rondon, et al. 2001. “A Founder Mutation in Presenilin 1 Causing Early-Onset Alzheimer Disease in Unrelated Caribbean Hispanic Families.” JAMA 286 (18): 2257–63. https://doi.org/10.1001/jama.286.18.2257.
Dumois-Petersen, Sofia, Martha P. Gallegos-Arreola, María T. Magaña-Torres, Francisco J. Perea-Díaz, John M. Ringman, and Luis E. Figuera. 2020. “Autosomal Dominant Early Onset Alzheimer’s Disease in the Mexican State of Jalisco: High Frequency of the Mutation PSEN1 c.1292C>A and Phenotypic Profile of Patients.” American Journal of Medical Genetics Part C: Seminars in Medical Genetics 184 (4): 1023–29. https://doi.org/10.1002/ajmg.c.31865.
Murrell, Jill, Bernardino Ghetti, Elizabeth Cochran, Miguel Angel Macias-Islas, Luis Medina, Arousiak Varpetian, Jeffrey L. Cummings, et al. 2006. “The A431E Mutation in PSEN1 Causing Familial Alzheimer’s Disease Originating in Jalisco State, Mexico: An Additional Fifteen Families.” Neurogenetics 7 (4): 277–79. https://doi.org/10.1007/s10048-006-0053-1.
Orozco-Barajas, Maribel, Yulisa Oropeza-Ruvalcaba, Alejandro A. Canales-Aguirre, and Victor J. Sánchez-González. 2022. “PSEN1 c.1292C<A Variant and Early-Onset Alzheimer’s Disease: A Scoping Review.” Frontiers in Aging Neuroscience 14: 860529. https://doi.org/10.3389/fnagi.2022.860529.
Parker, John, Tahseen Mozaffar, Ashlynn Messmore, Joshua L. Deignan, Virginia E. Kimonis, and John M. Ringman. 2019. “Homozygosity for the A431E Mutation in PSEN1 Presenting with a Relatively Aggressive Phenotype.” Neuroscience Letters 699 (April): 195–98. https://doi.org/10.1016/j.neulet.2019.01.047.
Rogaeva, Ekaterina. 2002. “The Solved and Unsolved Mysteries of the Genetics of Early-Onset Alzheimer’s Disease.” NeuroMolecular Medicine 2 (1): 1–10. https://doi.org/10.1385/NMM:2:1:01.
Santos-Mandujano, Rosalía A., Natalie S. Ryan, Lucía Chávez-Gutiérrez, Carmen Sánchez-Torres, and Marco Antonio Meraz-Ríos. 2020. “Clinical Association of White Matter Hyperintensities Localization in a Mexican Family with Spastic Paraparesis Carrying the PSEN1 A431E Mutation.” Journal of Alzheimer’s Disease: JAD 73 (3): 1075–83. https://doi.org/10.3233/JAD-190978.
Sepulveda-Falla, Diego, Markus Glatzel, and Francisco Lopera. 2012. “Phenotypic Profile of Early-Onset Familial Alzheimer’s Disease Caused by Presenilin-1 E280A Mutation.” Journal of Alzheimer’s Disease 32 (1): 1–12. https://doi.org/10.3233/JAD-2012-120907.
Soosman, Steffan K., Nelly Joseph-Mathurin, Meredith N. Braskie, Yvette M. Bordelon, David Wharton, Maria Casado, Giovanni Coppola, et al. 2016. “Widespread White Matter and Conduction Defects in PSEN1-Related Spastic Paraparesis.” Neurobiology of Aging 47 (November): 201–9. https://doi.org/10.1016/j.neurobiolaging.2016.07.030.
Yescas, Petra, Adriana Huertas-Vazquez, María Teresa Villarreal-Molina, Astrid Rasmussen, María Teresa Tusié-Luna, Marisol López, Samuel Canizales-Quinteros, and María Elisa Alonso. 2006. “Founder Effect for the Ala431Glu Mutation of the Presenilin 1 Gene Causing Early-Onset Alzheimer’s Disease in Mexican Families.” Neurogenetics 7 (3): 195–200. https://doi.org/10.1007/s10048-006-0043-3.
Round 2
Reviewer 1 Report
Comments and Suggestions for Authors
I don't have any additional comments.